# Pan-cancer molecular analysis of the RB tumor suppressor pathway

Erik S. Knudsen[1,2,3✉], Ram Nambiar[1,2], Spencer R. Rosario[1,4], Dominic J. Smiraglia [1,4], David W. Goodrich [1,5] & Agnieszka K. Witkiewicz[1,3,6✉]

The retinoblastoma tumor suppressor gene (*RB1*) plays a critical role in coordinating multiple pathways that impact cancer initiation, disease progression, and therapeutic responses. Here we probed molecular features associated with the RB-pathway across 31 tumor-types. While the RB-pathway has been purported to exhibit multiple mutually exclusive genetic events, only *RB1* alteration is mutually exclusive with deregulation of CDK4/6 activity. An ER+ breast cancer model with targeted *RB1* deletion was used to identify signatures of CDK4/6 activity and RB-dependency (CDK4/6-RB integrated signature). This signature was prognostic in tumor-types with gene expression features indicative of slower growth. Single copy loss on chromosome 13q encompassing the *RB1* locus is prevalent in many cancers, yielding reduced expression of multiple genes in cis, and is inversely related to the CDK4/6-RB integrated signature supporting a cause-effect relationship. Genes that are positively and inversely correlated with the CDK4/6-RB integrated signature define new tumor-specific pathways associated with RB-pathway activity.

[1] Roswell Park Comprehensive Cancer Center, Buffalo, NY 14203, USA. [2] Department of Molecular and Cellular Biology, Buffalo, USA. [3] Center for Personalized Medicine, Buffalo, USA. [4] Department of Genetics and Genomics, Buffalo, USA. [5] Department of Pharmacology and Therapeutics, Buffalo, USA. [6] Department of Pathology, Buffalo, USA. ✉email: erik.knudsen@roswellpark.org; agnieszka.witkiewicz@roswellpark.org

The *RB1* tumor suppressor was identified based on bi-allelic inactivation in retinoblastoma[1–3]. However, it has become clear that dysregulation of the RB protein pathway is a veritable hallmark of the cancer state. The RB-pathway is canonically described by oncogenes *CDK4*, *CDK6*, and *CCND1* and the tumor suppressors *RB1* and *CDKN2A*[2,4]. In this context, it has been proposed that these lesions are mutually exclusive and thus serve to define a simple linear pathway[5,6]. This network integrates mitogenic and oncogenic signaling pathways and plays critical roles in the etiology, progression, and therapeutic response in multiple different cancers[2,7].

Physiologically CDK4 and CDK6 activity is positively regulated by D-type cyclins that are responsive to proliferative signals that drive cell cycle progression[8–10]. The activity of these kinase complexes are kept in check by endogenous CDK4/6 inhibitors (e.g., CDKN2A) that limit inappropriate proliferation due to oncogenic signaling[11,12]. The principle target of this signaling network is the RB protein and related proteins encoded by the *RBL1* and *RBL2* genes[8,9,13,14]. The RB protein can function to repress a large transcriptional program of genes that are required for progression through S-phase, mitosis, and cytokinesis[15–17]. When dephosphorylated or hypophosphorylated RB is active in transcriptional repression; however, CDK4/6-mediated phosphorylation initiates the inactivation of RB and enables the expression of downstream genes that drive progression through the cell cycle and cell division[17–19]. Although much attention has been focused on cell cycle control mediated by RB, it is clear that RB-pathway impacts tumor metabolism, immunological features of the tumor microenvironment, and complex epigenetic states, often in a context-dependent fashion[2,3,20,21]. How the RB-pathway controls these context dependent features across human tumors remains poorly understood.

While *RB1* was the first tumor suppressor gene identified, there are a large number of questions that remain relative to biological functions that could promote targeted therapeutic strategies for the treatment of cancer[2,21]. At present, the therapeutic focus has been on leveraging CDK4/6 inhibition to activate RB and limit proliferation of tumor cells to delay disease progression[4,10]. In spite of success in ER+ breast cancer[22–24], this approach has not been as successful as anticipated across the large number of different tumor types where it has been tried. These clinical challenges suggest that an improved understanding of the circuitry involving the RB-pathway could be useful in either improving the utility of CDK4/6 inhibitors or advancing new approaches to target RB-pathway perturbation (e.g., via immunotherapy)[21,25]. For example, recent studies have suggested that the deregulation of cell cycle transitions upon *RB1* loss can represent a specific dependency on aurora kinases that can be targeted therapeutically[26,27]. In contrast to therapeutic exploitation of rapidly proliferating cancer cells, slow growing or dormant tumor cell populations can represent particular challenges since they are resistant to such therapies yet can seed recurrence and resistance to chemotherapy[28]. Therefore understanding how to eradicate biologically distinct forms of cancer is of clear significance.

Interrogating a large collection of molecular data from diverse tumor-types would be hypothesized to drive a clearer understanding of both the canonical features of RB-pathway cell cycle control as well as non-canonical, context dependent functions that may be leveraged for therapy. To this end, a pan-cancer molecular analysis of the RB-pathway was employed to probe genetic features across tumor types and gene expression relationships. This analysis sheds new light on genetic/gene expression interactions, how cell cycle regulatory networks related to *RB1* impact clinical outcomes, and has defined unique interactions that could represent new vulnerabilities for the treatment of cancer.

## Results

**Pan-cancer analysis of the core RB-pathway.** The core RB-pathway can be summarized by the proteins that define the kinase network that initiates RB protein phosphorylation (Fig. 1a). We used the TCGA pan-cancer collection analyzing 31 histological tumor types to interrogate the deregulation of this pathway across cancer (Supplementary Fig. 1). The genes encoding kinases (CDK4 and CDK6) and D-type cyclins (CCND1, CCND2, and CCND3) can have oncogenic function and are observed to be amplified or mutated in tumors (Fig. 1b and Supplementary Fig. 2). Conversely, loss of the endogenous CDK4/6 inhibitors (CDKN2A, CDKN2B, CDKN2C, and CDKN2D) or RB-family (*RB1*, *RBL2*, and *RBL1*) are also observed in specific tumor settings (Fig. 1b and Supplementary Fig. 2), in total the RB-pathway is subject to genetic perturbation in greater than 30% of tumors. Given the low frequency of some genetic events (e.g., RBL1 loss), we focused on the genes in the pathway most frequently altered (CDK4, CCND1, CDKN2A, and *RB1*). Surprisingly, and contrary to expectation, the genetic amplification of CDK4 and CCND1 are not mutually exclusive and appear to be co-occurring suggesting that these events coordinately contribute to disease (Fig. 1c and Supplementary Fig. 2). The principle CDK4/6 inhibitors lost in cancer are CDKN2A and CDKN2B. These two genes are located on chromosome 9p21 and therefore co-deletion is a common event (Fig. 1c and Supplementary Fig. 2). However, CDKN2A could be considered the critical tumor suppressor, as it is more commonly subject to single nucleotide variants and small insertions/deletions (Fig. 1b and Supplementary Fig. 2). Interestingly, genetic events targeting CDKN2A are not mutually exclusive with CDK4 amplification, and co-occurring with CCND1 amplification. This co-occurrence suggests these genetic events contribute additively to alter net CDK4/6 activity in tumors (Fig. 1c, d, and Supplementary Figs. 2, 3). In contrast, the *RB1* gene is mutually exclusive with the regulators of CDK4/6 activity (Fig. 1c, d, Table 1 and Supplementary Figs. 2–4). Most significantly CDKN2A loss and *RB1* loss are mutually exclusive in most cancers that lose these genes at a significant level (>5%), suggesting that the control of CDK4/6 activity and RB-status appear to define a single element of the pathway (Fig. 1e and Supplementary Figs. 2–4). The one exception to this relationship is in uterine corpus endometrial carcinoma (UCEC), where *RB1* mutations and CDKN2A mutations are co-occurring (Fig. 1e). Interestingly, these tumors are characterized as either microsatellite instable (MSI) or harboring POLE mutations; thus, co-occurrence could represent a "passenger phenomenon" in this specific tumor setting (Supplementary Fig. 5). Consistent with the mutual exclusive relationship between *RB1* and *CDKN2A* gene alterations, the genes are inversely expressed in most cancers exhibiting frequent loss/mutation of *RB1* (Fig. 1f and Supplementary Fig. 6). This reciprocal relationship was less evident in tumors that rarely exhibit *RB1* loss (e.g., colon cancer-COAD) or tumors where the RB-pathway is inactivated in veritably all tumors by the presence of HPV (e.g., cervical cancer-CESC) that yields RB inactivation via the E7 viral oncoprotein[29] (Fig. 1f). In HPV-positive CESC and squamous cell head and neck tumors (HNSC) there is highly-elevated expression of CDKN2A, indicative of RB inactivation by HPV (Supplementary Fig. 7). Similarly, in HNSC the majority of mutations targeting *CDKN2A* are in the HPV-negative tumor sub-type (Supplementary Fig. 7). Clustering analysis of the frequency of events targeting *CDK4*, *CCND1*, *CDKN2A*, and *RB1* indicates analyzed human cancers fall into five distinct groups separated largely by *RB1* status and the co-occurrence of *CCND1*, *CDK4*, and *CDKN2A* (Fig. 1g and Supplementary Fig. 8). For example, cluster 1 is dominated by CDK4 amplification with low frequency alteration in other RB-pathway genes (Fig. 1g). Unexpectedly, these clusters exhibit

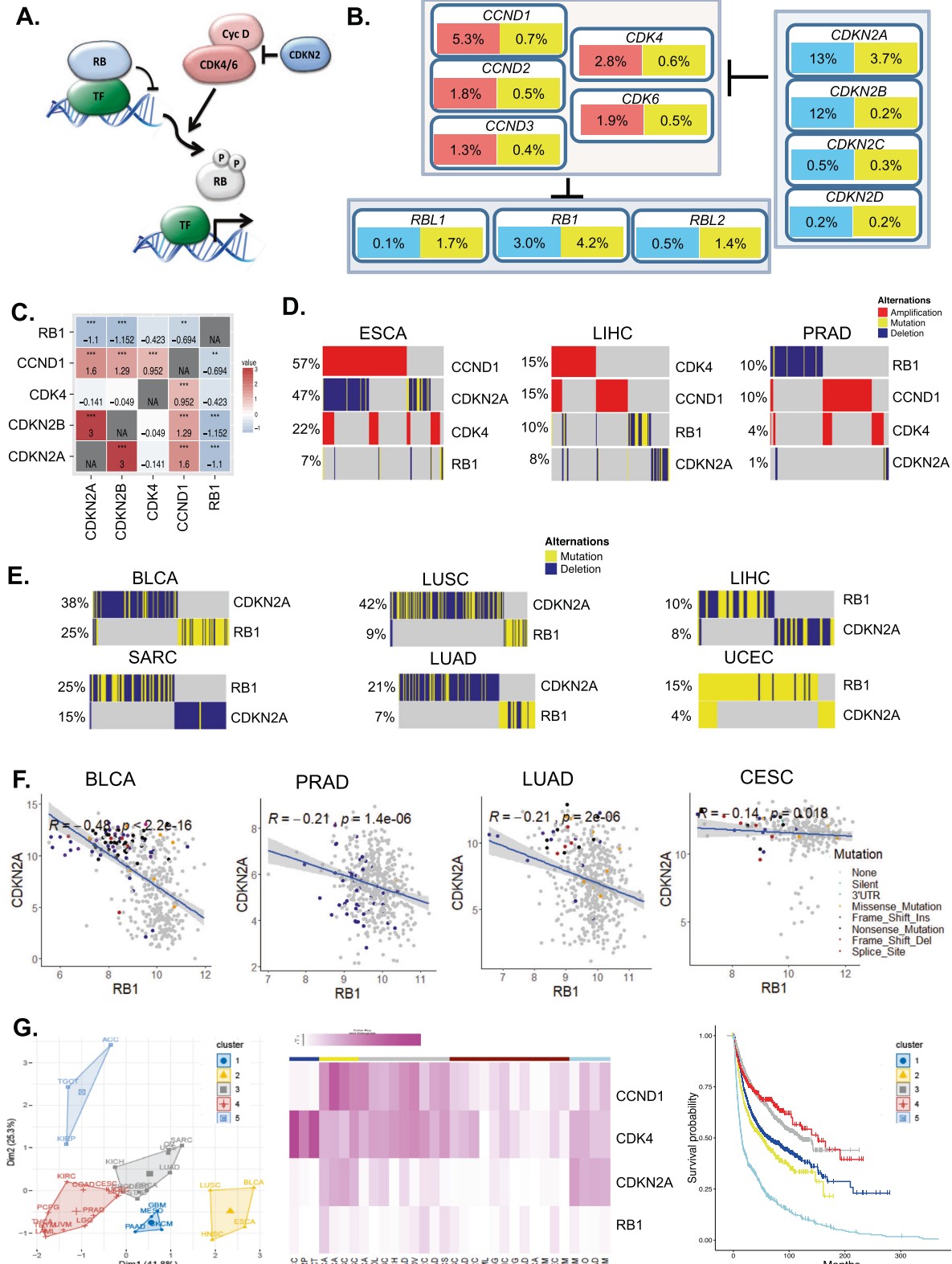

**Fig. 1 Analysis of the core RB-pathway. a** Generic depiction of the core RB-pathway. CDK4/6 activity is stimulated by D-type cyclins, which can be inhibited by members of the CDKN2 gene family. CDK4/6 activity converges on RB-family members to mediate phosphorylation and functional inactivation. **b** Summary of pan-cancer data indicating frequency of deletions (blue), amplifications (red), and mutations (yellow) affecting core pathway genes. **c** Odds ratio for co-occurrence (red) or mutual exclusivity (blue) are shown in the heatmap from all cancer cases (*$p < 0.05$, **$p < 0.01$, ***$p < 0.001$). **d** Oncoprints from ESCA, LIHC, and PRAD tumors are shown. **e** Oncoprints of tumors exhibiting relatively frequent RB loss/mutation. **f** Pearson correlation analysis was used to define the relationship between the expression of CDKN2A and *RB1*. The *R*-value and related *p*-value are shown, with the shading denoting the 95% confidence interval. The legend summarized genetic variations in the *RB1* gene. **g** The percentage of alterations in CCND1, CDK4, *RB1*, and CDKN2A were employed for K-means clustering across all tumor types. This approach yielded five clusters. The association of each cluster with disease-free survival is shown in the Kaplan–Meier plot.

**Table 1 The odds ratio and *p*-value for the relationship of CDKN2A and RB1 are provided for the indicated tumor types.**

| Cancer | Odds ratio | *p*-value |
|--------|-----------|-----------|
| BLCA | 0.10476 | 5.63E−14 |
| GBM | 0.17261 | 4.63E−08 |
| HNSC | 0.38812 | 0.03703 |
| LUAD | NA | 0.00021 |
| LUSC | 0.083624 | 5.16E−08 |
| PAAD | NA | 0.06059 |
| SARC | 0.070217 | 0.00030 |
| UCEC | 7.383507 | 1.40E−05 |

different disease-free survival (Fig. 1g), suggesting that different combinations of pathway perturbations could be associated with resultant clinical outcomes.

**Molecular analysis of chromosome 13q**. To interrogate the structural anomalies associated with *RB1* loss, we interrogated the copy number of genes along the q-arm of chromosome 13 (Fig. 2a and Supplementary Fig. 9). This analysis revealed that single copy loss of 13q occurs in a multitude of solid tumors, albeit occurring very infrequently in acute myeloid leukemia (LAML, colon adenocarcinoma (COAD), thyroid cancers (THYM, THCA), and uveal melanoma (UVM) (Supplementary Fig. 9). The distribution of the heterozygous events harbored three different chromosomal patterns. In the context of bladder and prostate cancer there is a frequent focal event that is focused over the *RB1* gene (Fig. 2a, b and S9). In multiple other tumor types (BRCA-breast, OV-ovarian, LIHC-liver hepatocellular carcinoma, LUAD-lung adenocarcinoma) there is a diversity of single copy deletions that ultimately coalesce over the genetic region around the *RB1* locus, but not BRCA2 which is located on the same chromosome arm (Fig. 2a, b and Supplementary Fig. 9). Lastly, there are tumor types wherein there is essentially no selection along the chromosome with the appearance of full loss of the chromosome arm (Fig. 2b and Supplementary Fig. 9).

To determine the significance of single copy loss of *RB1* we evaluated the relationship with the expression of *RB1*, and found that in the majority of tumor types single copy loss reduces *RB1* expression levels (Fig. 2c, d and Supplementary Fig. 10). Interestingly, the reduced expression occurs even in tumors where *RB1* activity is compromised via other mechanisms (e.g., cervical cancer) (Supplementary Fig. 10). The loss of single copy of *RB1* is associated with prognosis in specific tumor types, including those that rarely delete *RB1* (e.g., KIRC-clear cell kidney cancer) (Fig. 2c, d, and Supplementary Fig. 11). Interestingly, and distinct from deletion/mutation of *RB1*, the single copy loss of *RB1* is co-occurring with CDK4, CCND1, and CDKN2A genetic events (Supplementary Fig. 12), suggesting that there is the potential cooperation between reduced *RB1* gene dosage and CDK4/6 activity.

**Developing and deploying an RB-pathway activity signature**. Since *RB1* function can clearly be compromised by multiple events in addition to direct genetic loss in human cancers (e.g., HPV-E7 or CDK4/6 activity), we focused on developing a transcriptional tool to evaluate RB activity based on transcription that could be broadly deployed across tumors. To develop the "seed" for this signature, the ER+ breast cancer cell line MCF7 was treated with the CDK4/6 inhibitor palbociclib to mimic the action of CDKN2A and induction of RB activity (Fig. 3a). The RB-dependency of this signature was identified using an isogenic

model with RB-deleted. To ensure that this RB-pathway activity signature performed equally well in clinical specimens, we used METABRIC data from 2173 breast cancer cases to identify genes that maintained a high-degree of correlation in clinical cases (Supplementary Fig. 13). The resultant integrated signature of 182 genes (Supplementary Data 1) was reproducibly suppressed in clinical cases of ER+ breast cancer patients treated with palbociclib in the clinic[30] (Fig. 3b). Thus, we believe that this signature is a robust surrogate for the activation state of the RB-pathway at the cellular and tumor level.

The RB-pathway signature was employed to cluster all gene expression data in the TCGA pan-cancer data sets (Fig. 3c and Supplementary Fig. 14). The genes in the signature behaved in a highly co-regulated fashion in all tumors-types analyzed, even those with low levels of *RB1*-loss (Supplementary Fig. 14). High signature expression indicative of RB-pathway deregulation was prognostic in select tumor types (Fig. 3d and Supplementary Fig. 14). Interestingly, this prognostic activity was most significant in tumors that express relatively low levels of the CDK4/6-RB integrated signature, indicative of a more slow-growing disease. In tumors with high expression of the signature, the prognostic trends are generally not significant (Fig. 3e and Supplementary Fig. 14). In keeping with the biological features of cervical cancer (CESC), it expressed amongst the highest levels of the signature. Since many tumor types have distinct clinically relevant subtypes, considering subtype specific biology is important. In breast cancer, luminal and basal breast cancer forms were determined using PAM50, and as expected the luminal form of the disease expressed lower levels of the CDK4/6-RB integrated signature (Fig. 3f). Additionally, in this form of cancer the signature is prognostic, while not significantly associated with outcome in basal breast cancer (Fig. 3f). Similarly, HPV-positive head and neck cancers express significantly higher-levels of the signature as compared to HPV-negative subtype (Supplementary Fig. 15)

**Defining the effects of RB1 gene state on gene expression**. Using the genetic state of *RB1* locus in tumors, we defined genes that were differentially expressed in either the diploid vs. heterozygous or diploid vs. deleted genetic state. We focused on nine tumor types that exhibit >5% homozygous deletion frequency, and 24 tumors that exhibit >10% heterozygous loss. Significantly altered genes that occur in 66% of deleted cases (199 total: 27 downregulated, 162 upregulated), and 36% of heterozygous tumor types (141 total: 42 downregulated and 99 upregulated) were selected for analysis (Supplementary Datas 2 and 3). Strikingly, >90% of the down regulated genes in both the heterozygous and homozygous state were focused on 13q consistent with a *cis* relationship to *RB1* gene (Fig. 4a). An exception was the CCND1 gene, which was specifically repressed in tumors with homozygous deletion, consistent with the mutual exclusive relationship between *RB1* loss and CCND1 amplification (Fig. 4b and S16). Upregulated genes were enriched for cell cycle regulatory genes controlled by the E2F/FOXM1 transcription factors (Supplementary Fig. 17) and were significantly overlapping with both heterozygous loss and deletion of *RB1* (Fig. 4c). These genes were also highly enriched relative to the CDK4/6-RB integrated signature (Fig. 4d). Consistent with these findings in many tumor types, but not cervical cancer, the heterozygous loss of *RB1* was associated with elevated expression of the CDK4/6-RB integrated signature (Fig. 4e and Supplementary Fig. 18). Concordantly, the CDK4/6-RB integrated signature is often found to be upregulated in tumors that exhibit loss of CDKN2A or amplification of CCND1 and CDK4 (Supplementary Figs. 19–21).

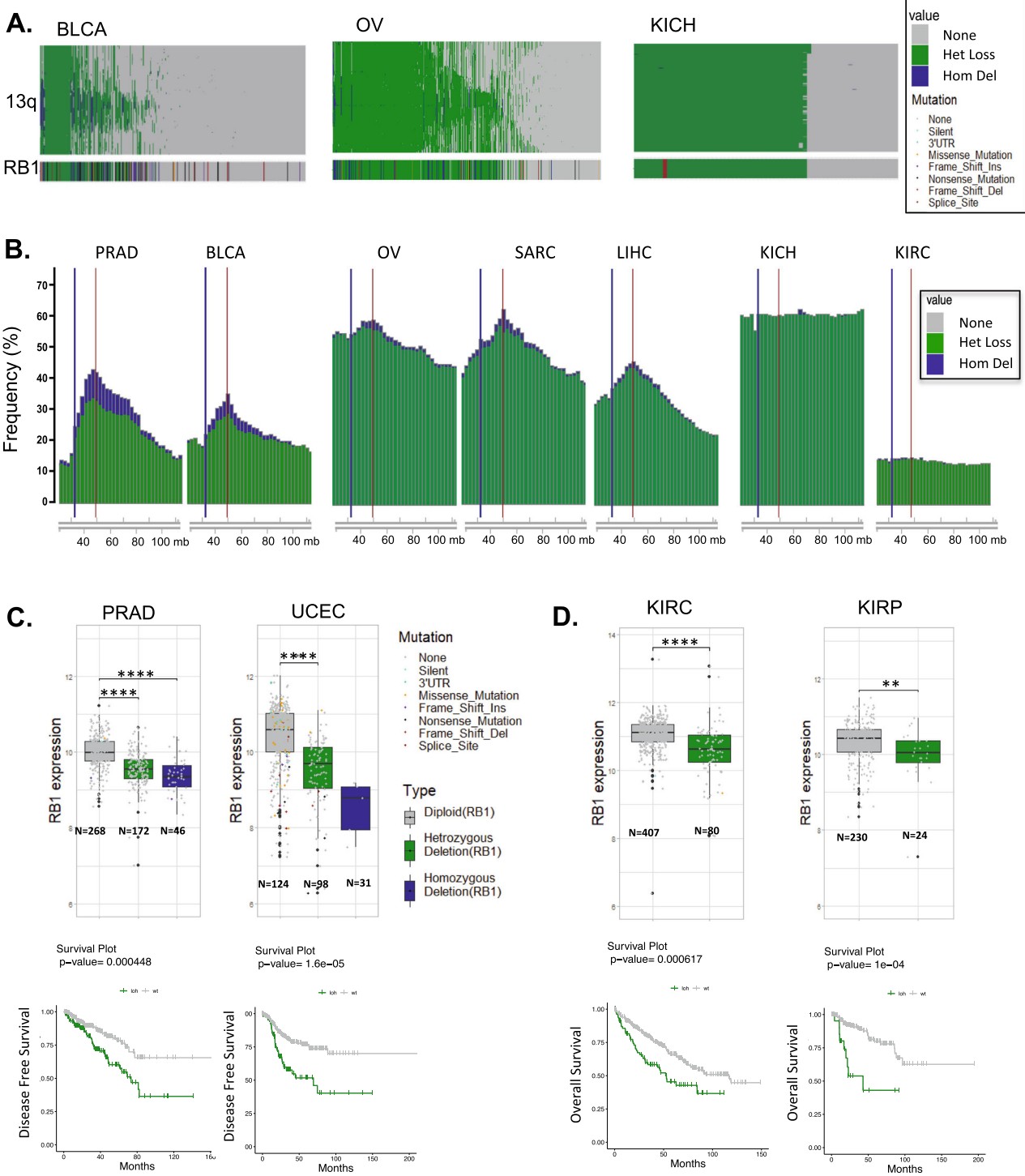

**Fig. 2 Perturbations of *RB1* locus in cancer. a** Copy number analysis of single copy loss (green) and deletion (blue) of genes in *cis* along chromosome arm 13q. Data is shown for BLCA, OV, and KICH, the *RB1* gene data is shown for reference. **b** Frequency plots of single copy (green) and deletion (blue) along 13q are shown for the indicated tumor types. The red line denotes the position of *RB1* and blue line denotes the position of BRCA2 on chromosome 13q (chromosomal location by nucleotide number is shown). **c** Analysis of *RB1* expression in tumors exhibiting diploid, heterozygous loss, or deletion of the *RB1* locus in PRAD and UCEC (Student's *t*-test two sided: **$p < 0.01$, ***$p < 0.001$,****$p < 0.0001$). Kaplan–Meier analysis of disease free survival by diploid vs. heterozygous loss of the *RB1* locus. Statistical analysis is by log-rank approach. **d** Analysis of *RB1* expression in tumors exhibiting diploid and heterozygous loss of the *RB1* locus KIRP and KIRC (Student's *t*-test two sided: **$p < 0.01$, ***$p < 0.001$). Kaplan–Meier analysis of overall survival stratified by diploid vs. heterozygous loss of *RB1* locus. Statistical analysis is by log-rank approach.

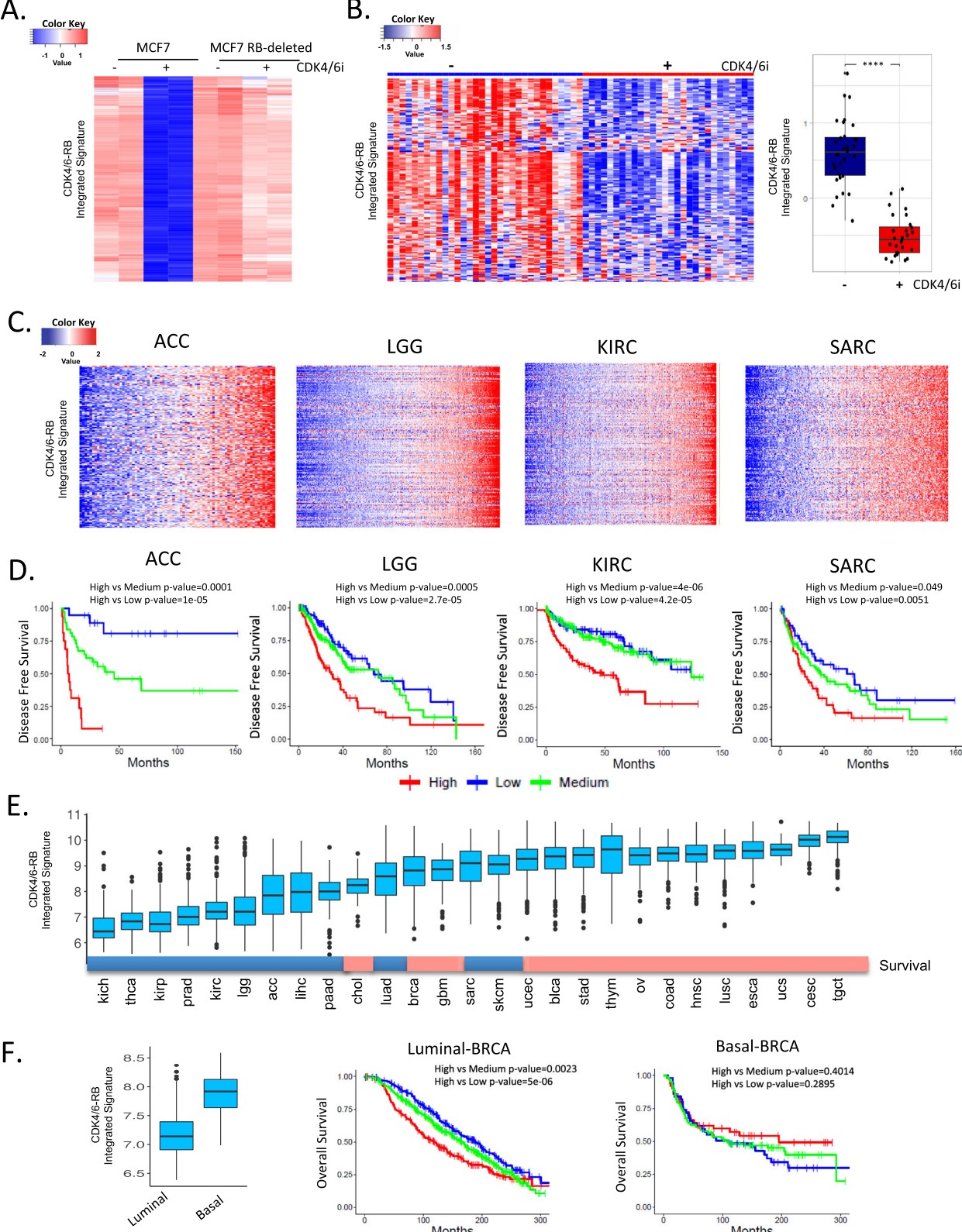

**Probing features of aggressive and indolent sub-types and novel vulnerabilities**. Since the analysis of 13q deletion defines a host of genes that are at that chromosomal location (i.e., regulated in *cis*) we utilized the CDK4/6-RB integrated signature to define genes that are positively and inversely correlated with deregulated pathway function. Bootstrapping was utilized to define significantly correlated genes in each tumor type (Supplementary

Data 4)[31]. Ranked gene set enrichment analysis (GSEA) defined relatively consistent enrichment for cell cycle nodes that are positively correlated as expected; however, there were a wide diversity of processes (e.g., lytic vacuole, fatty acid metabolism, and ion transport) that were inversely correlated (Fig. 5a and Supplementary Fig. 22). In spite of these differences relative to gene set enrichment, global analysis of the behavior of these

**Fig. 3 CDK4/6-RB integrated signature and prognosis. a** MCF7 cells and isogenic *RB1* deleted model were treated with CDK4/6 inhibitor (250 nM palbociclib) for 48 h. RNA sequencing was used to define transcriptional repression events that are RB dependent thereby linking CDK4/6 inhibition to RB-activation. **b** The CDK4/6-RB integrated signature was applied to data from the NeoPalaAnna trial and exhibited consistent repression in this clinical cohort of tumors treated with CDK4/6 inhibitors (pre-treatment $N = 32$, on treatment $N = 28$). Box plot shows difference in the integrated signature between pre and on-treatment samples (Student's *t*-test two sided: ****$p < 0.0001$) **c** Heatmaps illustrate that general behavior of the CDK4/6-RB integrated signature across several tumors types. **d** Kaplan–Meier analysis of the indicated tumor types stratified by CDK4/6-RB integrated signature. Disease-free survival is shown, and statistical analysis is by log-rank approach. **e** Pan-cancer relationship of the CDK4/6-RB integrated signature expression value. Color bar indicates tumors wherein high levels of the CDK4/6-RB integrated signature (indicative of high CDK4/6 activity or RB loss) are significantly associated with poor prognosis (blue) or not (red). **f** Relative expression of the CDK4/6-RB integrated signature in luminal ($N = 1175$) vs. basal ($N = 209$) breast cancer subtypes in the METABRIC data set (Student's *t*-test two side: $p < 0.001$). Kaplan–Meier analysis of luminal and basal breast cancer stratified by the CDK4/6-RB integrated signature. Statistical analysis is by log-rank approach.

collective genes across all tumor types revealed a general reciprocal relationship across essentially all tumor types (Fig. 5b). Clustering analysis revealed gene programs strongly correlated with the CDK4/6-RB integrated signature. While these clusters did include cell cycle regulated genes, there were a large number of genes that were associated with RNA-processing, protein synthesis, and nuclear pores (e.g., t-RNA export, cajal bodies, splicing, proteasome) (Fig. 5c) that are bound by E2F and Myc transcription factors (Supplementary Fig. 23). In general, the behavior of these genes was relatively consistent across tumor types, albeit cell cycle genes were the most strongly correlated (Fig. 5c). The negatively correlated genes were more variable in function and across tumor types, although they were statistically enriched for processes associated with immune function, metabolism, and features of signal transduction (Fig. 5d). To analyze potential causative effects, we investigated the behavior of the breast cancer bootstrap genes in MCF7 cells treated with palbociclib and found the directionality was consistent with the suppression of CDK4/6 activity (Fig. 5e). Many of the genes involved in processes observed to be positively correlated with CDK4/6 activity in TCGA data were, in fact, repressed with CDK4/6 inhibition (Fig. 5f). Similarly, treatment with CDK4/6 inhibition did induce genes that are inversely correlated with the CDK4/6-RB integrated signature (Fig. 5f). Together, these data suggest that there is a functional linkage between CDK4/6 activity and disparate gene expression programs suggesting possible new approaches for cancer treatment related to non-canonical features or CDK4/6-RB axis.

## Discussion

The RB tumor suppressor pathway has been extensively studied[1,3]; however, by analyzing a large collection of genetic and gene expression data across multiple tumor types it is possible to expand an understanding of genetic and biological complexities that reflect both tumor contexts and general principles. With the emergence of more targeted therapies that impinge on CDK4/6 and RB function understanding both canonical and less well-appreciated elements of the pathway will be important in advancing new therapeutic interventions.

The core RB-pathway understanding was built on mutual exclusive relationships that were observed in cell lines and surprisingly limited tumor analysis[5,32]. Such analysis pointed to exclusivity between CDK4 or CCND1 amplification and CDKN2A loss. In the larger genetic analysis it is clear that this is not the case, and many tumors harbor multiple genetic events that would be expected deregulate CDK4 activity. This observation would suggest that during tumor evolution there is continuing advantage to maximally deregulate CDK4/6, as supported by select functional studies[33]. In contrast *RB1* is mutually exclusive with multiple elements that drive CDK4 deregulation (i.e., CDKN2A loss, CCND1 amplification, and CDK4 amplification). These data and the reciprocal relationship between *RB1*

and CDKN2A expression strongly support the contention that *RB1* loss universally obviates the evolutionary advantage imparted by CDK4/6 deregulation. These data underscore the concept that RB-status should be generally considered as an exclusion criteria in the use of pharmaceutical agents that target CDK4/6 activity[34].

The evolutionary track of cancers can be inferred via the genetic features of tumor suppressor loss. In the context of *RB1*, there is frequent single copy loss along 13q in many tumor types. The structural changes and relationship to *RB1* deletion suggest different evolutionary forces at play in different tumors. In tumors such as prostate, bladder, and sarcoma the single copy loss is focused over the *RB1* locus with deletion occurring at reasonably high levels. Ostensibly these data reflect positive-selection associated with *RB1* loss and is consistent with the finding that *RB1* loss occurs more frequently in metastatic advanced cancers of this type (e.g., prostate and breast) and is associated with poor outcome[34,35]. Surprisingly, there are other tumor types where there is essentially no selection for deletion of *RB1* for example in kidney cancers or cervical cancer, even though there is substantial single copy loss. In the case of cervical cancer it is clear there is no evolutionary pressure to lose *RB1* due to HPV[36]. Why *RB1* loss does not occur in other cancers (e.g., kidney cancer) is less clear, however, since there is co-occurrence with deregulation of CDK4/6 activity the selective pressure may be limited. Similarly, since one copy loss appears to deregulate features of cell cycle perhaps there is little/no selective pressure to complete gene deletion. The functional analysis of single copy loss of *RB1* is very limited, although studies do suggest the loss of a single copy of *RB1* is sufficient to contribute to disease relevant phenotypes in mouse and human tumor models[37,38].

It is well known that a key feature downstream of the CDK4/6-RB pathway is transcriptional pathway coordinated by E2F and other transcriptional regulators[39,40]. Multiple gene expression signatures related to this pathway have been developed in different settings and models[19,41–45]. These signatures are well conserved and related to RB-dependent transcriptional repression across a host of tumor types. The CDK4/6-RB integrated signature developed herein behaves in a highly consistent fashion across all tumor types analyzed, suggesting that this gene expression module is essentially invariant. While this signature can have prognostic value, it is largely limited to those tumor types that have relatively low average signature value which would be inclusive of more indolent tumor types (e.g., prostate cancer and ER+ breast cancer). However, the signature has prognostic activity in tumors that are highly lethal (e.g., pancreatic cancer), albeit with lower proliferative indexes. The functional relationship between gene expression signatures linked to RB activity and actual *RB1* loss in cancer has been limited with disparate conclusions[44,46]. However, the vast majority of consistently repressed genes with *RB1* loss are downregulated in *cis*, both with single copy loss and *RB1* deletion. Those genes that are

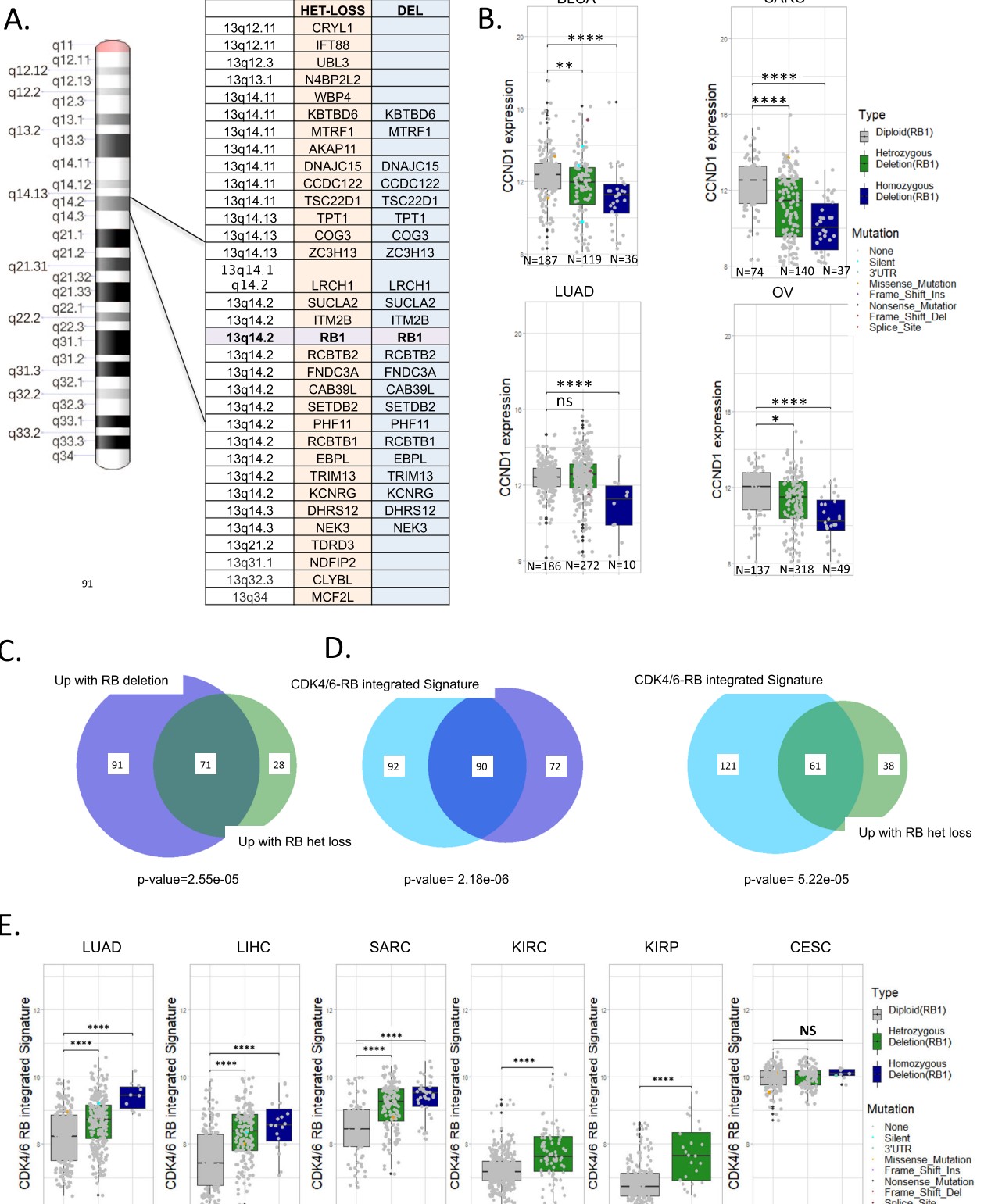

**Fig. 4 Association of *RB1* gene dosage with gene expression. a** Significantly altered genes between heterozygous loss and wild-type or deep deletion and wild-type were determined (1 logFC and *p* < 0.05). These gene lists were filtered for recurrence of >33% or >66% for heterozygosity and deletion respectively across different tumor types. >90% of the downregulated genes identified in this fashion were located in *cis* relative to *RB1* as shown. **b** Consistent with CCND1 amplification being mutually exclusive with *RB1* loss, *RB1* deletion was associated with lower expression levels of CCND1 (Student's *t*-test two sided:*p* < 0.05, **p* < 0.01, ***p* < 0.001, ****p* < 0.0001). **c** Relationship between consistently upregulated genes with heterozygous loss and deep deletion. Statistical significance was determined by hypergeometric test. **d** Intersection between upregulated genes with heterozygous loss or deep deletion with the CDK4/6-RB integrated signature. Statistical significance was determined by hypergeometric test. **e** Relationship between *RB1* gene dosage and the CDK4/6-RB integrated signature in select tumor types (Student's *t*-test two sided: ***p* < 0.001, **p* < 0.01, NS non-significant).

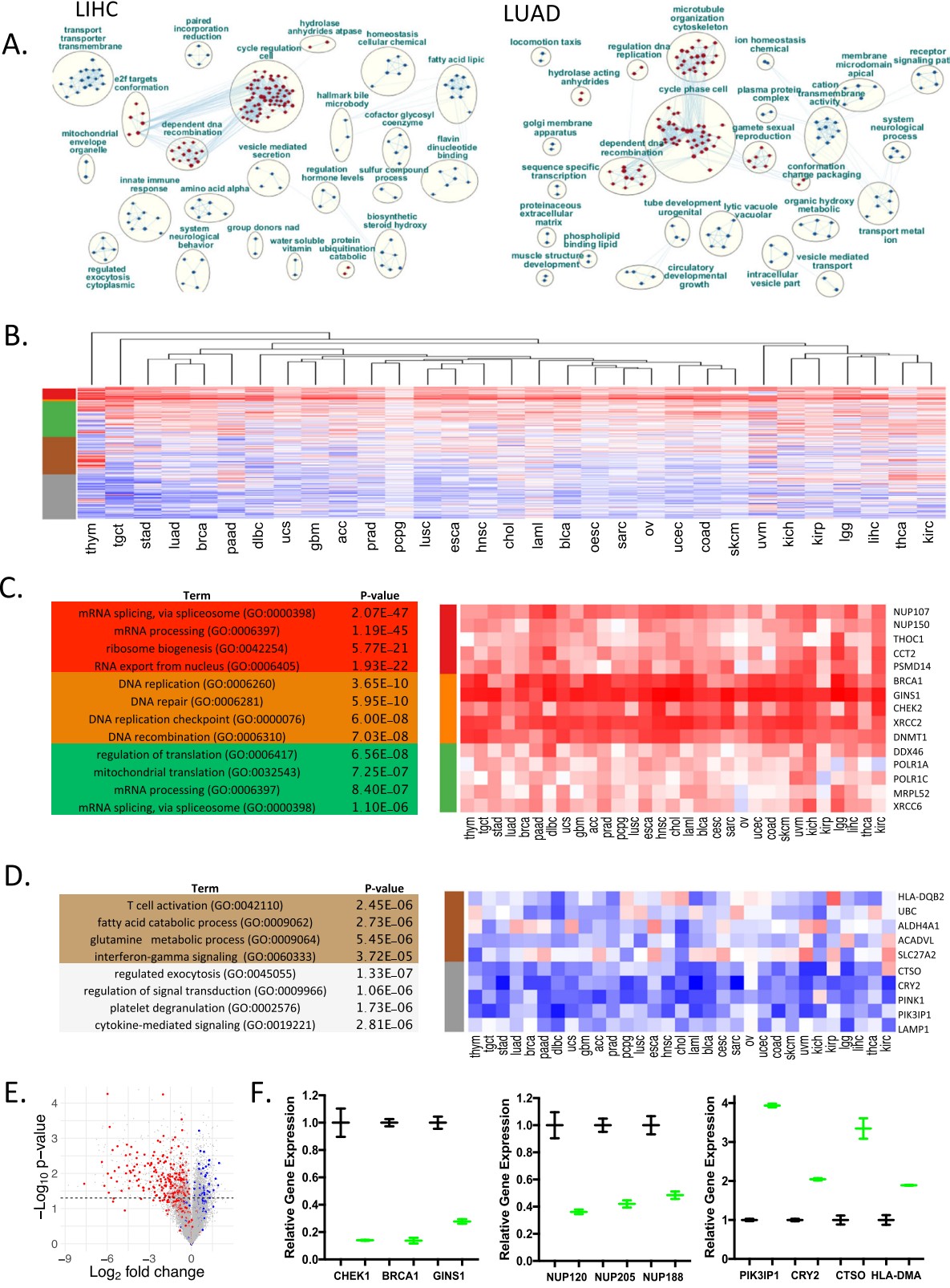

upregulated are largely related to cell cycle control/E2F activity and are coincident with the CDK4/6-RB integrated signature. These findings reinforce the concept that genetic aberrations targeting *RB1* do, in fact, drive deregulation of functionally derived signatures which are broadly manifest in cancer. Presumably, in tumor types such as LAML or COAD there are

pathway features distinct from RB loss that are driving the differential expression of these genes.

There is a large and expanding literature that CDK4/6 and RB activities are associated with biological features beyond cell cycle. Due to the limited genetic events targeting RB, gene expression analysis provides a more statistically-powerful approach to mine

**Fig. 5 Defining pathway features beyond cell cycle. a** To define features beyond cell cycle control bootstrapping was performed to define genes that were positively and inversely correlated with the CDK4/6-RB integrated signature. Ranked gene set enrichment coupled with reactome pathway analysis generated key nodes of regulation for each tumor types. **b** K-means clustering of the correlation coefficients yielded three predominantly positively correlated clusters, and two inversely correlated clusters. **c** Gene ontology and transcription factor enrichment analysis for the genes in the positively correlated clusters was performed in ENRICHR. Top enriched gene ontologies and transcription factor binding sites are shown. Heatmap showing selected genes from each of the clusters. **d** Gene ontology and transcription factor enrichment analysis for the genes in the negatively correlated clusters was performed in ENRICHR. Top enriched gene ontologies and transcription factor binding sites are shown. Heatmap showing selected genes from each of the clusters. **e** Relationship of correlated genes for breast cancer relative to gene expression analysis from MCF7 cells treated with CDK4/6 inhibitors (red = positively correlated, blue = negatively correlated). **f** Expression levels of select genes in the positively and negatively correlated expression groups in DMSO control (black) or palbociclib treated (green) groups. The mean and standard deviation of the gene expression are shown. Statistical analysis was determined by Student's t-test: CHEK1 $p = 0.014$, BRCA1 $p = 0.002$, GINS1 $p = 0.004$, NUP120 $p = 0.02$, NUP205 $p = 0.009$, NUP188 $p = 0.019$ PIK3IP1 $p < 0.001$, CRY2 $p = 0.002$, CTO $p = 0.014$ HLA-DMA $p = 0.019$.

for features that are associated with pathway status. Utilizing this approach a large number of genes positively and inversely correlated with the CDK4/6-RB integrated signature can be identified. Interestingly, while there are some differences between tumor types the directionality of these correlations is surprisingly consistent. By analyzing the features that are positively correlated it suggests vulnerabilities that could be particularly relevant in targeting cells with cell cycle deregulation. These analyses encompass cell cycle genes (e.g., AURKA and CHEK1) that have been shown to be preferentially actionable against *RB1* deficient tumor models[26,27]. However, these data also illustrate that these same tumors are preferentially engaged in translation, splicing, proteasome function, and other cellular functions that are likely critical for facilitating rapid cell division (e.g., synthesis of nuclear pore components). Therefore, these tumors could be selectively sensitive to agents that target these processes (e.g., proteasome inhibition or unfolded protein stress). This concept has been tested in limited contexts evaluating either genetic interactions (e.g., between *RB1* deletion and targeting RNA processing)[47] or drug sensitivities specific to RB-deficient tumors[48]. Conversely, we identify a host of processes that would appear to be more engaged in indolent tumors or those that have forced out of the cell cycle therapeutically. This includes genes involved in immunological responses that have been functionally associated with response to immune checkpoint inhibition in tumor models exposed to CDK4/6 inhibitors[25,49]. Whether other processes (e.g., mitophagy or metabolism) could be selectively targeted in such tumor cell populations will need to be determined given the profound alterations in genes associated with these processes. However, there is evidence that dormant tumors are susceptible to metabolic therapies or targeting autophagy[50,51]. Together, this analysis provides targets that are impacted by the RB-pathway that could yield clinically relevant vulnerabilities.

## Methods

**Data retrieval, oncoprints, mutual exclusivity, and correlation**. All the TCGA datasets were acquired from cBioportal from the PanCancer Atlas Study https://www.cbioportal.org/datasets. These data consist of 31 histologically distinct tumor types. Oncoprint graphs were generated using the ComplexHeatmap function in R. The correlation between *RB1* and CDKN2A gene expression was calculated using Pearson correlation coefficient, where "R" indicates the correlation coefficient with the associated p-value. Mutual exclusivity was determined using log odds ratio with the associated p-value. The NeoPalaAnna data were downloaded from the gene expression omnibus (GSE93204) and Metabric data were downloaded from cbioportal.org.

**Focused analysis of genes on chromosome 13q**. All the genes present on Chromosome-arm 13q were obtained using the biomaRt package from Bioconductor. The information regarding the copy number variance files were downloaded from TCGA Pan-cancer analysis for each tumor type. The frequencies of heterozygous or homozygous deletion were mapped in the *cis* gene order. The gene expression levels of *RB1* were stratified by deep deletion, single copy loss and wild-type gene status. Box plots show the median, inter-quartile range and minimum/maximum data points for all figures. A two-sided, Student's t-test was

performed in R to find the significance as represented in the boxplot. Similar analysis was performed with survival data for each cancer types. Kaplan–Meier Survival analysis between wild-type and heterozygous gene loss was performed using the "survival" package in R. Statistical significance was determined by a log-rank test.

**Definition and application of the CDK4/6 integrated signature**. MCF7 cells were obtained from the ATCC before 2010. The presence of mycoplasma was routinely assessed (every 3–6 months) using DAPI staining methodology on fixed cells. The cells retain estrogen receptor and harbor the morphology of MCF7 cells; the STR testing was performed on Feb 14, 2020 to confirm identity. The CRISPR-mediated deletion was achieved using guide sequences designed to target exon 2 of RB1 (CACCGAGAGAGAGAGCTTGGTTAACTT). A CAS9 expressing plasmid and RB1 target plasmid were co-transfected and single cell clones were developed. The targeting of RB was confirmed by immunoblotting and immunofluorescence as we have published. Parental MCF7 cells and those rendered RB-deleted by CRISPR-mediated gene targeting were treated with vehicle control or 250 nM palbociclib for 48 h. RNA recovered from duplicate biological samples were subjected to Illumina-based RNA sequencing (data is deposited in GEO under the manuscript title). The resultant Fastq files were processed using HTSeq. The RNA seq count files were then normalized by counts per million (CPM) using the edgeR package in R. Those genes that are downregulated in MCF7 cells treated with palbociclib were identified (LogFC less than or equal to −1, $p < 0.05$). These genes were filtered with identically treated RB deleted MCF7 cells, to define genes significantly repressed in an RB-dependent fashion ($n = 362$ genes). These genes were applied to the METABRIC Gene expression dataset to evaluate performance in clinical tumor specimens. Genes with a maximum correlation value less than the mean of the maximum correlation values, were filtered out and the remaining genes were used as CDK4/6 integrated signature totaling ($n = 182$ genes).

The CDK4/6 integrated signature genes were applied to the gene expression data for each cancer type and normalized across all cases. The pan-cancer cases were then clustered based on their average signature expression across all genes in the CDK4/6-RB integrated signature. The cases were then stratified into three groups, lowest 25%, medium 50% and highest 25%. Kaplan–Meier survival analysis was performed on these groups.

**Identification of genes associated with *RB1* single copy loss and deep deletion**. Venn diagram shows the overlap between most frequently occurring genes that exhibit LOH across all Cancer types, most frequently occurring genes that exhibit Deep deletion across all cancer types and genes in the cell cycle signature. Only those genes that exhibit LOH in more than 33% of the cancer types and genes that exhibit deep deletion in more than 66% of the cancer types were considered. The pairwise significance between these three gene sets were calculated using a hypergeometric test.

**Identification of gene programs associated with the CDK4/6-RB integrated signature**. The Pearson correlation between each gene, with respect to the average gene expression of CDK4/6-RB integrated signature genes, was calculated for each cancer type. For an unbiased analysis, a bootstrapping algorithm was used by sampling genes 10,000 times with replacement. From this analysis, the lower limit and upper limit in a 95% confidence interval was used as cutoff with genes whose Pearson correlation coefficient is lesser than the lower limit being negatively correlated, and genes whose Pearson correlation coefficient is greater than the upper limit being positively correlated with CDK4/6 integrated signature genes. Only those genes whose correlation with CDK4/6 integrated signature genes are significant (p-value <0.05) were considered for this analysis. All the bootstrapped genes across all cancer types were hierarchically clustered based on their corresponding correlation coefficients to produce five distinct clusters. Gene ontology and transcription factor enrichments were determined using ENRICHR (https://amp.pharm.mssm.edu/Enrichr/).

For the analysis of the MCF7 dataset, the log fold change and the p-value were calculated using a standard two tailed student $t$-test of CDK4/6 inhibitor treated relative to DMSO control to generate a volcano plot. The corresponding genes that were highly negatively correlated (represented as blue) and highly positively correlated (represented as red) with the cell cycle genes, from the bootstrapped gene lists for breast cancer (BRCA).

**Gene set enrichment analysis.** For each cancer type, the highly positively correlated and highly negatively correlated genes were combined and their corresponding weights were calculated using the formula: $W(i) = CC(i) * (-\log10(p\_val(i)))$ Where, for each gene i, the corresponding weight $W(i)$ is the product of the Pearson correlation coefficient of the gene ($CC(i)$) and corresponding $p$ value($p\_val(i)$). Next, gene set enrichment analysis (GSEA) using the pre-ranked gene set along with the corresponding weights was performed to find Gene Ontology (GO) terms. Networks showing the relationship between GO terms were generated using the java-based application Cytoscape program called Enrichment Mapper. Common GO terms were then clustered and annotated.

**Statistics and reproducibility.** The number of samples is provided in the figure or employed the totality of the tumor population. Statistical approaches and replicates are described in the methods above, or in the figure legends.

**Reporting summary.** Further information on research design is available in the Nature Research Reporting Summary linked to this article.

## Data availability

Any data that is not available in the supplement or through the links provided above will be provided by the corresponding author (ESK) on request.

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

## Acknowledgements
The authors would like to thank their colleagues for thought-provoking discussion on this study. The work was supported by grants to A.K.W. (CA211878) and E.S.K. (CA188650 and CA127387). D.W.G. is supported by grants (CA234162 and CA207757).

## Author contributions
Study conception and design was performed by E.S.K., D.W.G., R.N., S.R.R., D.J.S., and A.K.W. Acquisition of data was performed by R.N., S.R.R., and E.S.K. Analysis and interpretation of data was performed by E.S.K., D.W.G., R.N., S.R.R., D.J.S., and A.K.W. Drafting of manuscript was performed by E.S.K., D.W.G., and A.K.W. Financial support for the study was provided by E.S.K. and A.K.W.

## Competing interests
The authors have no competing financial or non-financial interests associated with the present study. Erik Knudsen is an Editorial Board Member for *Communications Biology*, but was not involved in the editorial review of, nor the decision to publish this article.
