## [Peer Review File · Communications Biology]

Reviewers' comments:

Reviewer #1 (Remarks to the Author):

Knudsen et al. performed a deep and comprehensive study of genes involved in RB/CD4/6 pathway, and suggested new treatment options based on this pathway. Below are my comments:

Co-occurrence or mutual exclusivity for genes involved in RB1 pathway:

It would be more informative if the authors can show these relationships across major cancer types with altered RB1 pathway (Figure 1 and Figure S2).

Can the authors list the MSI status for UCEC in Figure 1E? I suspect these samples with high mutational burden are MSI-high samples.

Relationship of RB1 to virus:

The authors state that "The reciprocal relationship was less evidence in tumors that rare exhibit RB loss.... Or tumors where the RB1-pathway is inactivated in veritably ... by the presence of HPV (e.g. cervical cancer-CESC)".

Change "all tumor" to "all tumors".

Can the authors look into HPV-negative CESC to investigate if it is unique to HPV-positive CESC? Also, Can they validate the finding in HPV-positive HNSC? They can find the HPV status for CESC and HNSC tumors in Nature Communications, 2015 and Scientific Reports 2016.

CDK4/6-RB integrated signature:

Can the authors cite the supplementary table in the main text when they first mentioned the 182 genes on page 8?

Also, What is the concordance of these genes with known genes in RB1 pathway?

It would be helpful if the authors have signature-related score system to rank samples when they use them to quantify RB1-pathway activity,.

Can the authors clarify what different colors stand for in Fig. 3E?

Pathway enrichment analysis:

It would be better if the sidebar colors can be consistent with the colors of different pathways in the left of Figure 5C. It seems that the authors only list one gene for each pathway in Fig. 5C and D, what is the selected criteria for these genes?

I can not find Fig. 6G and Fig. 6H.

Clinical relevance:

The authors state that the current study provides new non-canonical genes in RB1 pathway for treatment option. Can they be more specific on that statement? Like which genes were discovered in the study, not in the previous studies.

Reviewer #2 (Remarks to the Author):

Synopsis:

In the current work by Knudsen *et al.*, the Authors carried out a pan-cancer analysis of the RB-pathway leveraging publicly available mutational and gene expression data from TCGA. The premise of the work relies on the fact that although genetic alterations affecting *RB1* and associated genes e.g. *CDK4/6*, *CCND1*, *CDKN2A/B/C/D* in the canonical RB1-pathway are well characterized, a unifying theme seems to be missing. There is a cancer specific context in the alterations affecting the components of the RB1-pathway with for instance ACCs displaying mostly amplifications in *CDK4* whilst in contrast, GBMs harbor most often homozygous deletions of *CDKN2A*. The work of Knudsen *et al.* is timely. In fact, although the community has been quick to embrace the results of large phase III trials of *CDK4/6* inhibitors e.g. PALOMA3 (PMID: 26030518) to the point that it has become standard of care in advanced stage breast cancers, there are (1) baseline resistant cases even in cancers where *CDK4/6* inhibitors have had high success rates, (2) acquired resistance and (3) cancer types which are altogether refractory. Although points (1) and (2) are not the subject of the current work, surely on account of the cancer specific context of RB1-pathway alterations, point (3) which this Reviewer had some difficulty in disentangling from the introduction is the centerpiece of this manuscript. This Reviewer is positive about the work of Knudsen *et al.* but is of the opinion that several issues need to be addressed. The following list of critiques is provided as hopefully constructive feedback that the Authors might capitalize upon to improve their manuscript.

Major comments:

1/ In the introduction between lines #45 and #84, although the Authors provide a succinct overview of the literature which to some extent alludes to the results which are being presented later in the manuscript, this Reviewer failed to appreciate the last paragraph between lines #67 and #84 which falls short of providing clear rationales. As such, the introduction reads as the premise to an exploratory analysis, the depth of which is left to the better judgement of the Readers. This Reviewer is of the opinion that if the Authors could clearly state what are the primary/ secondary objectives, incidental findings falling into the category of interesting results would be better articulated whilst not distracting from the main message of the work.

2/ In Figure 1F and Figure S5, the different colors of the dots are not explained although one may guess that it refers to patients/ samples with a genetic alteration in any one of *CDKN2A*, *RB1* or both. Can the Authors provide a key in the legend or a description in the caption? Similarly, it seems that the confidence intervals have not been defined. These are scatterplots showing the association between the expression of two genes whilst "Pearson's correlation" is one of many ways of measuring the strength of this association. In both Figures, the scatterplots seem to have been rasterized and the title added afterwards. In Figure S5 in particular, the individual data points and axis labels have been expanded without constraining the proportions. It is possible for the Authors to render this using *ggplot* in the R library *ggplot2* as it is difficult to appreciate the panels as they are currently displayed. Lastly, although for some cancer types e.g. BRCA, the trend seems to be linear and a least-square regression seems appropriate, in other types of cancers, the variance in *CDKN2A* expression either does not seem to be explained by that of *RB1* or is actually non-linear. The p-values are significant owing mostly to the number of data points but the *Rs* are generally low. As the Authors correctly argue though, this may be due to the cancer specific context (lines #117 and #118) In the opinion of this Reviewer, this hypothesis is testable beyond shading of data points through the use of multiple regression using the *glm* framework with mutations/ copy number alterations in *CDKN2A* and *RB1* and possibly interactions of the two as covariates. Can the Authors elaborate?

3/ In Figure 1G, although it is a decent effort and unless the Authors can provide the rationale substantiated by literature, this Reviewer is of the opinion that survival data from different cancer

types cannot be pooled together for the purpose of Kaplan-Meier analysis. Rather the approach taken by the Authors in Figure 2D and Figure 3D to stratify by cancer type is the correct one.

4/ In Figure 5A-D, it is difficult to understand the point made by the Authors. It could be due to the fact that differential expression analysis and Gene Set Enrichment Analysis (GSEA) in particular are hard to represent graphically. This Reviewer suggests that the Authors represent pathway analysis as in Jiménez-Sánchez *et al.* (PMID: 28841418) which in that particular article is based on single sample GSEA but should be easier for group based differential expression.

5/ Figure S6 captures from a high level the similarities and differences of cancer types through a principal component analysis of the frequencies of alterations in specific components of the RB1-pathway. Although it is not specified how the cancer types were clustered, one could achieve similar results though a *k*-means or to the very least a hierarchical clustering. This Reviewer highly appreciated how well this Figure captures the problem at hand. Would it be possible for the Authors to elaborate and move this Figure as a panel in one of the main Figures of the manuscript?

Minor comments:

1/ At places, the manuscript is uneven in the use of English and contains typographical errors.

2/ In Figure 1D-E and Figure S3, there genes are ordered by frequency of alterations in decreasing order. It is difficult to compare the different cancer types side-by-side. Can the Authors choose a given order and remain consistent throughout the manuscript or at least in the Figures cited above?

3/ In Figure 2B, the chromosome ideograms are rasterized and it is difficult to visualize the cytogenetic bands since none of them have been labelled. There is a multitude of R/ Bioconductor libraries which can render chromosome ideograms. Please see Gviz (10.18129/B9.bioc.Gviz) or copynumber (10.18129/B9.bioc.copynumber) for instance.

Reviewer #3 (Remarks to the Author):

This paper describes a pan-cancer largely exploratory analysis of RB-pathway by assessing genomic and transcriptomic data. The main findings of the study are:

- mutual exclusivity is only observed between RB1 loss and genomic events that result in CDK4/6 deregulation;
- single copy RB1 loss is prevalent in most cancers and results in reduced RB1 expression;
- the established CDK4/6-RB integrated signature can be prognostic for a subset of cancer types (higher signature expression with worse survival).

The authors also explored pathways that are correlated with the established signature, hence identifying new potential therapeutic avenues. The analysis and results of this study will be of interest to a broad community of cancer researchers.

Major comments

1. The discussion is clearly written and most conclusions are not overstated, which is appropriate for such exploratory analysis. However, the results were difficult to follow because limited details of the analysis were provided:
 - a. Number of samples in each cancer type is not listed (can be added to Fig S1);
 - b. Number of genes is not listed for any histograms;
 - c. Number of samples not listed for any figure;
 - d. A lot of figure legends are unclear or not provided (ie Fig 3A – what is value?, Fig 5C/D and Fig S11 – no legend; Fig S12 – labels are unreadable; Fig S13 – “deepskyblue” is not an appropriate label);
 - e. Figure descriptions are often unclear (ie Fig 5A – is this all pathways identified across all tumour types?);
 - f. Most supplementary figures require a more detailed description;

- g. Since only a subset of cancer types is shown in main figures, an explanation of why these cancer types are shown would be useful, especially because different cancer types are shown in each figure;
 - h. Cancer types are presented in different order in most supplementary figures;
 - i. No explanation for how genes are selected for Fig 5C,D and F;
 - j. No descriptions or labels are provided for supplementary tables making them uninterpretable.
2. In the 'co-occurrence/mutual exclusivity analysis' authors rightly point out that tumours with high-mutation burden can have passenger events that can look like co-occurrence. Multiple statistical methods have been developed to account for tumour variability, for example, WExT (Leiserson et al Bioinformatics 2016) or DISCOVER (Canisius et al Genome biology 2016). Authors should either use one of such methods in parallel with odds ratios, or soften some of their conclusions in this section, for example regarding CDKN2A events co-occurring with CCND1 amplification.
 3. Figure 2D shows survival plots for a subset of cancers. The authors are showing a mix of Disease-Free Survival and Overall Survival plots. Mixing survival types is not a standard accepted practice, and should be avoided unless a specific reason is given. In Figure 3D the survival type is not specified.
 4. In the discussion, authors noted that some biological observations could be explained by cancer types being more indolent than others. It would be useful if the authors included a figure to categorically break down cancers by survival. Accordingly, using this or an alternative categorical grouping of tumour types (i.e. by frequency of RB-pathway events) would help with interpreting main and supplementary figures.
 5. Could not find a description of how MCF7 cells were cultured or RNAseq was generated. Was this done in a different study?
 6. Could not find a description of how NeoPalaAnna trial data was obtained.

Minor comments:

1. Fig 4D: CDK4/6-RB integrated signature has 178 genes; while in methods it has 183 genes.
2. RB1-pathway and RB-pathway are used interchangeably.
3. Line 111: what is a significant level of gene loss in cancer types? Was there a specific cut-off that was used?
4. Lines 122, 124 and 128: instead of Fig 1E and 1F; Fig 1F and 1G should be listed, respectively.
5. Fig S4: Can CDKN2A be added?
6. Fig S9: KIRC cancer type is listed twice and KIRP is missing.
7. Unclear of how CDK4/6-RB integrated signature score is calculated.
8. Fig 3B. perhaps the differences in CDK4/6-RB integrated signature scores should be accessed between untreated and treated cases.

RESPONSE: The authors thank the reviewers for their rigorous review of the submitted manuscript. We have extensively edited the manuscript to address veritably all of the points raised and have extensively proof-read the manuscript. Below is a point-by-point response to the reviewers' comments, wherein the reviewer comments are shown verbatim and our response is provided in a blue font.

Reviewer #1 (Remarks to the Author):

Knudsen et al. performed a deep and comprehensive study of genes involved in RB/CD4/6 pathway, and suggested new treatment options based on this pathway. Below are my comments:

1) Co-occurrence or mutual exclusivity for genes involved in RB1 pathway:

It would be more informative if the authors can show these relationships across major cancer types with altered RB1 pathway (Figure 1 and Figure S2).

This analysis has been performed for multiple tumor types that exhibit frequent RB loss, this is shown in the revised Figure S3.

Can the authors list the MSI status for UCEC in Figure 1E? I suspect these samples with high mutational burden are MSI-high samples.

As shown in the revised Figure S5, the vast majority of RB1 and CDKN2A point mutations occur in the MSI or POLE mutant UCEC tumors. These data indicate that the coordinate mutation of RB1 and CDKN2A is a consequence of the high mutation burden in such tumors.

2) Relationship of RB1 to virus:

The authors state that "The reciprocal relationship was less evidence in tumors that rare exhibit RB loss..... Or tumors where the RB1-pathway is inactivated in veritably ... by the presence of HPV (e.g. cervical cancer-CESC)".

Change "all tumor" to "all tumors".

We regret the typographical error pointed out, and have extensively reviewed the manuscript for grammatical errors.

Can the authors look into HPV-negative CESC to investigate if it is unique to HPV-positive CESC? Also,

Can they validate the finding in HPV-positive HNSC? They can find the HPV status for CESC and HNSC tumors in Nature Communications, 2015 and Scientific Reports 2016.

As suggested, we stratified the expression level of CDKN2A by HPV-status in HNSC and CESC tumor samples. Those tumors which are HPV-positive express high-levels of CDKN2A (new Figure S7). Furthermore, the loss of RB and CDKN2A are largely restricted to HNSC tumors that are HPV-negative (new Figure S7).

3)CDK4/6-RB integrated signature:

Can the authors cite the supplementary table in the main text when they first mentioned the 182 genes on page 8?

The data table (Supplemental data table 1) is now referenced in the appropriate location.

Also, What is the concordance of these genes with known genes in RB1 pathway?

In addition to our analysis of the genetic changes of RB1 on the CDK4/6-RB integrated signature, we also evaluate the impact of CCND1, CDK4 amplification events and CDKN2A loss. As expected in certain tumor types where these events are prevalent, there is an association with the gene expression signature (revised Figures S19-21).

It would be helpful if the authors have signature-related score system to rank samples when they use them to quantify RB1-pathway activity.

As shown in figure 3E there is clearly tumor selective diversity of the signature expression; therefore, we believe employing the signature in individual tumor types (without a pre-determined score) is more appropriate.

Can the authors clarify what different colors stand for in Fig. 3E?

The color-coding pertains to a significant impact of the CDK4/6-RB integrated signature on disease-free survival. This is clarified in more detail in the revised figure legend.

4) Pathway enrichment analysis:

It would be better if the sidebar colors can be consistent with the colors of different pathways in the left of Figure 5C. It seems that the authors only list one gene for each pathway in Fig. 5C and D, what is the selected criteria for these genes?

The genes from the heatmap in Fig 5C are selected from Gene Ontologies shown in the table. The Table shows the top 4 gene ontology terms associated with genes in each clusters.

The Table colors in Fig 5C have been changed to match cluster color of the heatmap as suggested.

5)Clinical relevance:

The authors state that the current study provides new non-canonical genes in RB1 pathway for treatment option. Can they be more specific on that statement? Like which genes were discovered in the study, not in the previous studies.

The study indicates the presence of pathways that correlate with RB cell cycle deregulation including splicing and translation, suggesting that such tumors could be more sensitive to such interventions. In fact, functional studies have suggested this intersection and are now discussed in the revised manuscript. Relative to indolent tumors there is induction of immunological and metabolic pathways that could represent new targets for intervention. For example PINK1, Tp53INP1, TP53INP2, PARK2 are all genes identified here that are not conventionally associated with cell division. These points are discussed more explicitly in the revised discussion section.

Reviewer #2 (Remarks to the Author):

Synopsis:

In the current work by Knudsen *et al.*, the Authors carried out a pan-cancer analysis of the RB-pathway leveraging publicly available mutational and gene expression data from TCGA. The premise of the work relies on the fact that although genetic alterations affecting *RB1* and associated genes e.g. *CDK4/6*, *CCND1*, *CDKN2A/B/C/D* in the canonical RB1-pathway are well characterized, a unifying theme seems to be missing. There is a cancer specific context in the alterations affecting the components of the RB1-pathway with for instance ACCs displaying mostly amplifications in *CDK4* whilst in contrast, GBMs harbor most often homozygous deletions of *CDKN2A*. The work of Knudsen *et al.* is timely. In fact, although the community has been quick to embrace the results of large phase III trials of *CDK4/6* inhibitors e.g. PALOMA3 (PMID: 26030518) to the point that it has become standard of care in advanced stage breast cancers, there are (1) baseline resistant cases even in cancers where *CDK4/6* inhibitors have had high success rates, (2) acquired resistance and (3) cancer types which are altogether refractory. Although points (1) and (2) are not the subject of the current work, surely on account of the cancer specific context of RB1-pathway alterations, point (3) which this Reviewer had some difficulty in disentangling from the introduction is the centerpiece of this manuscript. This Reviewer is positive about the work of Knudsen *et al.* but is of the opinion that several issues need to be addressed. The following list of critiques is provided as hopefully constructive feedback that the Authors might capitalize upon to improve their manuscript.

Major comments:

In the introduction between lines #45 and #84, although the Authors provide a succinct overview of the literature which to some extent alludes to the results which are being presented later in the manuscript, this Reviewer failed to appreciate the last paragraph between lines #67 and #84 which falls short of providing clear rationales. As such, the introduction reads as the premise to an exploratory analysis, the depth of which is left to the better judgement of the Readers. This Reviewer is of the opinion that if the Authors could clearly state what are the primary/ secondary objectives, incidental findings falling into the category of interesting results would be better articulated whilst not distracting from the main message of the work.

We thank the reviewer for their positive view of the study, we have modified the introduction to better address the objectives of this work and the underlying hypotheses being interrogated.

In Figure 1F and Figure S5, the different colors of the dots are not explained although one may guess that it refers to patients/ samples with a genetic alteration in any one of *CDKN2A*, *RB1* or both. Can the Authors provide a key in the legend or a description in the caption?

We regret the oversight the caption has been added to indicate the color-coding.

Similarly, it seems that the confidence intervals have not been defined. These are scatterplots showing the association between the expression of two genes whilst “Pearson’s correlation” is one of many ways of measuring the strength of this association. In both Figures, the scatterplots seem to have been rasterized and the title added afterwards. In Figure S5 in particular, the individual data points and axis labels have been expanded without constraining the proportions. It is possible for the Authors to render this using *ggplot* in the R library *ggplot2* as it is difficult to appreciate the panels as they are currently displayed. Lastly, although for some cancer types e.g. BRCA, the trend seems to be linear and a least-square regression seems appropriate, in other types of cancers, the variance in *CDKN2A* expression either does not seem to be explained by that of *RB1* or is actually non-linear. The p-values are significant owing mostly to the number of data points but the *Rs* are generally low. As the Authors correctly argue though, this may be due to the cancer specific context (lines #117 and #118) In the opinion of this Reviewer, this hypothesis is testable beyond shading of data points through the use of multiple regression using the *glm* framework with mutations/ copy number alterations in *CDKN2A* and *RB1* and possibly interactions of the two as covariates. Can the Authors elaborate?

Again, we regret the oversight the confidence interval information has been added to the figure legend and we have attempted to ensure that all data is appropriately scaled. We used ggPlot2 as suggested to determine more complex non-linear relationships between CDKN2A and RB1 as suggested. Because there are multiple mechanisms through which RB protein function is limited (beyond genetic alterations testable herein) the use of ggPlot2 would indicate that the relationship is dependent on additional factors. This is discussed in the revised text; however, in most tumor types RB mutation/deletion is associated with increased CDKN2A expression.

In Figure 1G, although it is a decent effort and unless the Authors can provide the rationale substantiated by literature, this Reviewer is of the opinion that survival data from different cancer types cannot be pooled together for the purpose of Kaplan-Meier analysis. Rather the approach taken by the Authors in Figure 2D and Figure 3D to stratify by cancer type is the correct one.

We don’t dispute that the analysis is a bit unorthodox; however, we were very surprised that the frequency and distribution of lesions in the RB-pathway could be of prognostic significance. In keeping with the suggestions of the same reviewer (below), we move the principle component analysis to the main figure and indicate the caveat of the survival analysis in the revised text.

4) In Figure 5A-D, it is difficult to understand the point made by the Authors. It could be due to the fact that differential expression analysis and Gene Set Enrichment Analysis (GSEA) in particular are hard to represent graphically. This Reviewer suggests that the Authors represent pathway analysis as in Jiménez-Sánchez *et al.* (PMID: 28841418) which in that particular article is based on single sample GSEA, but should be easier for group based differential expression.

The points of Figures 5A-D was to define relationships beyond cell cycle that could be related to the CDK4/6-RB pathway. We used the approach of correlation-bootstrapping to obtain significantly correlated genes for each tumor type. These genes were used for ranked GSEA analysis to define pathway that are positively negative correlated with the CDK4/6-RB integrated signature (Fig 5A) In the context of Figures 5B-D, this is analysis of the defined genes from individual tumor types that are interrogated across all tumors. To us it was surprising how well conserved were the gene expression behaviors across cancer (Figure 5B) and the functional features that expand well beyond the cell cycle (Fig 5C-D).

5) Figure S6 captures from a high level the similarities and differences of cancer types through a principal component analysis of the frequencies of alterations in specific components of the RB1-pathway. Although it is not specified how the cancer types were clustered, one could achieve similar results though a *k*-means or to the very least a hierarchical clustering. This Reviewer highly appreciated how well this Figure captures the problem at hand. Would it be possible for the Authors to elaborate and move this Figure as a panel in one of the main Figures of the manuscript?

We have moved this figure to the main figures as suggested and downplay the survival data in accordance with the point raised above.

Minor comments:

1) At places, the manuscript is uneven in the use of English and contains typographical errors.

We have extensively proof-read the revised manuscript, by multiple native English speakers.

2) In Figure 1D-E and Figure S3, there genes are ordered by frequency of alterations in decreasing order. It is difficult to compare the different cancer types side-by-side. Can the Authors choose a given order and remain consistent throughout the manuscript or at least in the Figures cited above?

We are using the common convention for oncprints, since the number of variables is relatively limited keeping the order by frequency illustrates that different tumors have different predominant mechanisms of pathway aberration.

3) In Figure 2B, the chromosome ideograms are rasterized and it is difficult to visualize the cytogenetic bands since none of them have been labelled. There is a multitude of R/ Bioconductor libraries which can render chromosome ideograms. Please see Gviz (10.18129/B9.bioc.Gviz) or copynumber (10.18129/B9.bioc.copynumber) for instance.

As suggested, we rendered the data with precise chromosome location indicated and the specific location of the BRCA2 and RB1 denoted with a vertical line. Irrespective of the approach the chromosome banding is essentially impossible to see, so we are just illustrating the coordinates in the revised version of the figure.

Reviewer #3 (Remarks to the Author):

This paper describes a pan-cancer largely exploratory analysis of RB-pathway by assessing genomic and transcriptomic data. The main findings of the study are:

- mutual exclusivity is only observed between RB1 loss and genomic events that result in CDK4/6 deregulation;
- single copy RB1 loss is prevalent in most cancers and results in reduced RB1 expression;
- the established CDK4/6-RB integrated signature can be prognostic for a subset of cancer types (higher signature expression with worse survival).

The authors also explored pathways that are correlated with the established signature, hence identifying new potential therapeutic avenues. The analysis and results of this study will be of interest to a broad community of cancer researchers.

Major comments

1. The discussion is clearly written and most conclusions are not overstated, which is appropriate for such exploratory analysis. However, the results were difficult to follow because limited details of the analysis were provided:

a. Number of samples in each cancer type is not listed (can be added to Fig S1);

We have added the information regarding samples numbers for each tumor type in the revised Fig S1. In general the full complement of TCGA cases is being utilized for each analysis; however, in specific instances with stratification different numbers of cases are being employed this is now specifically shown in the figure or the accompanying legend. Additionally, we now indicate the number of samples in the non-TCGA studies employed.

b. Number of genes is not listed for any histograms;

We regret the oversight we have added the number of genes to the figure legends for each heatmap or histogram being employed where it is not explicitly indicated (e.g. the CDK4/6-RB integrated signature always contains the same genes)

c. Number of samples not listed for any figure;

In all cases we used the full TCGA pan-cancer sample collection as summarized in the revised Fig S1. In cases where stratification was employed (e.g. by HPV-status), the number of samples with stratification is provided in the figure or accompanying legend. For non-TCGA data the number of samples is provided in each figure legend.

d. A lot of figure legends are unclear or not provided (ie Fig 3A – what is value?, Fig 5C/D and Fig S11 – no legend; Fig S12 – labels are unreadable; Fig S13 – “deepskyblue” is not an appropriate label);

We regret these oversights. With the number of figures involved we have enlisted several members of the group to review the figures. We have also extensively modified the figure legends and enhanced the graphics as suggested.

e. Figure descriptions are often unclear (ie Fig 5A – is this all pathways identified across all tumour types?);

We regret these oversight, the figure legends have been extensively expanded

f. Most supplementary figures require a more detailed description;

We regret these oversight, the figure legends have been extensively expanded

g. Since only a subset of cancer types is shown in main figures, an explanation of why these cancer types are shown would be useful, especially because different cancer types are shown in each figure;

In general, the tumor types that clearly illustrate the point are being shown in the main text. For example, in Figure 1E we show tumors for which there is either mutual exclusivity or co-occurrence. All the other tumor types are shown in the supplement.

h. Cancer types are presented in different order in most supplementary figures;

In the revised manuscript we have maintained the order of the tumors as reported by TCGA's naming scheme that is summarized in Figure S1. This ordering scheme is not always self-evident (e.g. LGG is before Breast Cancer, because LGG represents "Brain Low Grade Glioma"). In the supplemental data the listing is left to right then top to bottom.

i. No explanation for how genes are selected for Fig 5C,D and F;

These are subset of genes for each of the clusters shown. The genes were selected to illustrate the terms from the gene-ontology.

j. No descriptions or labels are provided for supplementary tables making them uninterpretable.

The supplementary data tables have been annotated as suggested

2. In the 'co-occurrence/mutual exclusivity analysis' authors rightly point out that tumours with high-mutation burden can have passenger events that can look like co-occurrence. Multiple statistical methods have been developed to account for tumour variability, for example, WEXT (Leiserson et al Bioinformatics 2016) or DISCOVER (Canisius et al Genome biology 2016). Authors should either use one of such methods in parallel with odds ratios, or soften some of their conclusions in this section, for

example regarding CDKN2A events co-occurring with CCND1 amplification.

We agree that softening the conclusions is best, which we have done in the revised text.

3. Figure 2D shows survival plots for a subset of cancers. The authors are showing a mix of Disease-Free Survival and Overall Survival plots. Mixing survival types is not a standard accepted practice, and should be avoided unless a specific reason is given. In Figure 3D the survival type is not specified.

Fig 2D shows the most significant disease-free and overall survival differences. We agree with the reviewer and have re-organized the data to show disease-free and overall survival in different panels.

4. In the discussion, authors noted that some biological observations could be explained by cancer types being more indolent than others. It would be useful if the authors included a figure to categorically break down cancers by survival. Accordingly, using this or an alternative categorical grouping of tumour types (i.e. by frequency of RB-pathway events) would help with interpreting main and supplementary figures.

The use of the term “indolent” here is perhaps not ideal and should rather be “rapidly dividing”. For example it is difficult to claim pancreatic cancer is indolent, although the tumors have a lower proliferation rate than other cancers. As suggested, we have determined the DFS and OS for all tumor types to integrate with Figure 3E.

5. Could not find a description of how MCF7 cells were cultured or RNAseq was generated. Was this done in a different study?

This information have been added to the revised methods section of the manuscript and the data is being deposited in GEO.

6. Could not find a description of how NeoPalaAnna trial data was obtained.

We regret the oversight the data was downloaded from gene expression omnibus (GSE93204)

Minor comments:

1. Fig 4D: CDK4/6-RB integrated signature has 178 genes; while in methods it has 183 genes.

The signature is composed of 182 genes, we regret the error in the text.

2. RB1-pathway and RB-pathway are used interchangeably.

We use the terminology RB1 for the retinoblastoma tumor suppressor gene. However, since the pathway is related to protein function we have changed to RB-pathway throughout the text. We also have appropriately demarcated genes with italics.

3. Line 111: what is a significant level of gene loss in cancer types? Was there a specific cut-off that was used?

Yes the cutoff is >3%

4. Lines 122, 124 and 128: instead of Fig 1E and 1F; Fig 1F and 1G should be listed, respectively.

We regret the oversight, we have corrected the error and also extensively cross-referenced the text with the figures.

5. Fig S4: Can CDKN2A be added?

These data have been added along with the MSI and POLE status in the revised figure S5.

6. Fig S9: KIRC cancer type is listed twice and KIRP is missing.

We regret the oversight we have extensively checked the supplemental figures for any data omission or duplication.

7. Unclear of how CDK4/6-RB integrated signature score is calculated.

For the ranking of cases we simply apply the average signature level as a continuous variable this is now indicated in the revised methods section.

8. Fig 3B. perhaps the differences in CDK4/6-RB integrated signature scores should be assessed between untreated and treated cases.

In the revised Figure 3, we include the box-plot of the CDK4/6-RB integrated signature from the NeoPalaAnna trial as suggested.

REVIEWERS' COMMENTS:

Reviewer #1 (Remarks to the Author):

The authors addressed most of my past comments. However, there are still some left. Additionally, some new results created questions, as indicated below:

What are the meanings of the percentages in Figures S5 and S7?

The authors state that RB1 loss is restricted to HPV-negative samples. However, it seems to be not true from Figure S7. What is the evidence they used to draw this conclusion?

How did the authors get the HPV status for HNSC and CESC, and MSI/POLE status for UCEC tumors?

As addressed in my previous comments, how did the authors select one gene for each pathway in Figure 5C?

Reviewer #3 (Remarks to the Author):

The authors have adequately addressed majority of the reviewers' comments, significantly improving data presentation and readability of the manuscript.

One area that still requires further clarification is the methodology section for the "Definition and application of the CDK4/6 integrated signature" experiment. The authors added that they generated a CRISPR-modified cell line and later produced RNAseq data, however more detail is needed on:

1. When and where the cell line (MCF7) was obtained from, as well as if and when it was last tested for mycoplasma and authenticated. This is part of Nature Research requirements (detail currently not provided in the "reporting summary").
2. CRISPR/Cas9 methodology used, i.e. gRNA sequence, vector/virus delivery method.
3. RNA extraction, library generation and sequencing parameters.
4. How RNAseq data was analysed.

Dear Editor,

We thank the reviewers for their important suggestions. We have addressed the points as recommended below (our response in blue text). These changes are demarcated in the “marked version” of the manuscript using track changes. Relative to the MCF7 cells, we have updated the scientific reporting information as requested by the reviewer. We are excited by the prospect of publication in Communications Biology. Thank you for your help in this process.

Reviewer #1 (Remarks to the Author):

The authors addressed most of my past comments. However, there are still some left. Additionally, some new results created questions, as indicated below:

What are the meanings of the percentages in Figures S5 and S7?

The percentage in Fig S5 and S7 denotes the percentage of cases with amplification/deletion. This information is added to the revised supplement figure legends.

The authors state that RB1 loss is restricted to HPV-negative samples. However, it seems to be not true from Figure S7. What is the evidence they used to draw this conclusion?

We regret the mistake, we have changed the text to more accurately reflect the data.

How did the authors get the HPV status for HNSC and CESC, and MSI/POLE status for UCEC tumors?

The Subtypes are provided in the TCGA Clinical data that is part of the standard downloaded data.

As addressed in my previous comments, how did the authors select one gene for each pathway in Figure 5C?

The genes were selected from the gene list based for the Gene-ontology. We could have shown more genes, but limited to a single gene just to make the text legible in the figure.

Reviewer #3 (Remarks to the Author):

The authors have adequately addressed majority of the reviewers' comments, significantly improving data presentation and readability of the manuscript.

One area that still requires further clarification is the methodology section for the "Definition and application of the CDK4/6 integrated signature" experiment. The authors added that they generated a CRISPR-modified cell line and later produced RNAseq data, however more detail is needed on:

1. When and where the cell line (MCF7) was obtained from, as well as if and when it was last tested for mycoplasma and authenticated. This is part of Nature Research requirements (detail currently not provided in the "reporting summary").

This information is added to the revised Methods section and are incorporated in the revised reporting summary.

2. CRISPR/Cas9 methodology used, i.e. gRNA sequence, vector/virus delivery method.

The MCF7 RB-deleted cell line has been previously described. However, we have added a brief description of the approach employed.

3. RNA extraction, library generation and sequencing parameters.

This information has been added to the revised methods section.

4. How RNAseq data was analysed.

The MCF7 RNASeq count files were obtained from the fastq files through an established RNA seq pipeline. The RNA seq count files were then normalized by counts per million(CPM) using the edgeR package in R